# Low-Temperature Rheological Properties and Microscopic Characterization of Asphalt Rubbers Containing Heterogeneous Crumb Rubbers

**DOI:** 10.3390/ma13184120

**Published:** 2020-09-16

**Authors:** Mingfeng Chang, Yixing Zhang, Jianzhong Pei, Jiupeng Zhang, Min Wang, Fugui Ha

**Affiliations:** 1Department of Materials Science and Engineering, School of Materials Science and Engineering, Chang’an University, 2nd Ring Rd South East Section, Xi’an 710064, China; 2020131065@chd.edu.cn (Y.Z.); 2017902514@chd.edu.cn (M.W.); 2018901708@chd.edu.cn (F.H.); 2Key Laboratory for Special Area Highway Engineering of Ministry of Education, Chang’an University, 2nd Ring Rd South East Section, Xi’an 710064, China; pei@chd.edu.cn (J.P.); zhjiupeng@chd.edu.cn (J.Z.)

**Keywords:** asphalt rubber, heterogeneous crumb rubbers, rheological property, microscopic characterization, linear viscoelastic material functions

## Abstract

Asphalt rubbers mixed with untreated and plasticized crumb rubbers and a compounding coupling agent were investigated in this study. The low-temperature rheological properties of asphalt rubbers at different aging levels were tested using a dynamic shear rheometer (DSR). An interconversion between linear viscoelastic material functions was used to obtain converted evaluation indexes for the asphalt rubbers at low temperatures. Lastly, the physicochemical characteristics and the microscopic morphology of the asphalt rubbers were evaluated using Fourier transform infrared spectroscopy (FTIR) and scanning electron microscopy (SEM), respectively. In conclusion, the storage moduli of the asphalt rubbers containing heterogeneous crumb rubbers increased with the plasticized crumb rubber content and the aging level. The converted relaxation moduli were consistent with the change trend of the storage moduli, and the relaxation rate decreased as the plasticized crumb rubber content and the aging level increased. The process of mixing the base asphalt with crumb and plasticized crumb rubbers was physical blending, and the effect of aging on the absorption peak change of asphalt rubber with plasticized crumb rubbers was less than that of asphalt rubber with ordinary crumb rubbers. Aging deteriorated the blending between the crumb rubber and the base asphalt, and a distinct interface appeared between the crumb rubber and the base asphalt. The particle cores of the plasticized crumb rubber in the asphalt rubber were difficult to maintain. Furthermore, as the plasticized crumb rubber content increased, more fine particles stripped off the plasticized crumb rubber after aging.

## 1. Introduction

Crumb rubber prepared by waste rubber tires can be effectively combined with a base asphalt in asphalt rubber pavement, which saves resources and presents an environmentally sustainable treatment for hundreds of millions of waste tires produced each year. There are great differences in the compositions of crumb rubber due to the different composition of waste tires, which affects the performance of asphalt rubber [1,2]. Moreover, the poor compatibility between crumb rubber and asphalt in a traditional asphalt rubber results in low performance. However, plasticization of untreated crumb rubbers cleaves existing cross-links. The resulting short molecular chains can move and unfold easily in the base asphalt, which improves blending with the base asphalt.

An increasing number of studies have been carried out on asphalt rubber for use as a road material [3,4,5]. Multistress creep recovery and frequency sweep tests were used to determine the rheological properties, i.e., the high-temperature and fatigue properties, of asphalt rubber mixed with a warm agent and a liquid antistripping agent, as well as of SBS (styrene-butadiene-styrene) biorubber asphalt [6,7,8]. Zhang et al. evaluated the high-temperature performance of asphalt rubber using dynamic shear rheometry and a multistress creep recovery test [9]. Wang et al. used temperature sweep and multistress creep recovery tests to evaluate the high-temperature performance of asphalt rubber treated with bio-oil [10]. Pei et al. investigated the high-temperature rheological properties of asphalt rubber containing crumb rubber that had been treated with bio-oil and zinc stearate [11,12]. Singh et al. added different crumb rubber contents to a recycled asphalt binder and used multistress creep recovery and linear amplitude sweep tests to evaluate the high-temperature rut-resistance of the asphalt rubber [13,14]. Xiao et al. found that the high-temperature rheological properties of asphalt rubber were significantly improved by the addition of plasma-treated crumb rubber [15]. Ameri et al. used a linear amplitude sweep test to determine the viscoelastic fatigue characteristics of asphalts modified by crumb rubber and SBS [16]. Yu et al. comprehensively investigated the fatigue performance of asphalt rubber using the fatigue factor index, a linear amplitude sweep test, an indirect tensile fatigue test and a four-point bending beam test [17].

The sensitivity of asphalt rubber to temperature was evaluated using a bending beam rheometer (BBR) and a dynamic shear rheometer (DSR) to determine the low-temperature evaluation index and the mechanism by which particles affect the high-temperature rheological properties of asphalt rubber [18,19]. He et al. evaluated the rheological properties of a warm mix Sasobit asphalt rubber at high, medium and low temperatures [20,21]. Puga et al. investigated the low-temperature performance of asphalt rubbers prepared with different asphalt binders, crumb rubber contents and coupling agents and determined the impact of the terminal blend (TB) molecular components on the low-temperature performance of asphalt rubber [22,23,24]. Furthermore, as an additive used for many years, Vestanamer was a thermoplastic polymer with a unique molecular structure, which achieved the cross-linking of crumb rubber and asphalt, and improved the road performance of asphalt rubber and asphalt rubber pavement [25,26,27,28].

Summarizing and analyzing the existing research on asphalt rubber, it can be concluded that the current research trends mainly focus on testing the high-temperature rheological properties of asphalt rubber, and the high-temperature performance is improved after adding crumb rubber. But there are few studies on low-temperature performance. In view of the plasticized crumb rubber containing more small molecular composition and the cross-linking effect of Vestanamer, studies on the low-temperature performance of asphalt rubber prepared by mixing untreated and plasticized crumb rubber with Vestanamer have not yet been reported. However, low-temperature cracking is a well-known defect in asphalt pavement. The same problem is encountered with asphalt rubber prepared using untreated and plasticized crumb rubbers. Thus, the low-temperature rheological properties of asphalt rubber prepared using heterogeneous crumb rubbers needs to be investigated together with the microscopic characteristics of the asphalt rubber for different aging levels to elucidate the mechanisms involved in modification and aging. The objective of this study was to investigate the impact of heterogeneous crumb rubbers on the low-temperature rheological properties and physiochemical characteristics of aged asphalt rubbers. The aging mechanism was elucidated by determining the microstructure of asphalt rubber prepared with heterogeneous crumb rubbers.

## 2. Materials and Methods

### 2.1. Materials

#### 2.1.1. Base Asphalt

The base asphalt used in this study was SK 90# with PG 58 −22 °C produced in Korea (South Korea SK Group, Seoul, Korea). The measured basic property indexes and compositions of SK 90# are listed in Table 1 and Table 2. The Fourier transform infrared spectrum (Nicolet iS5, Thermo Fisher Scientific, Waltham, MA, USA) of the base asphalt and SEM ((JSM-6010LA, JEOL Ltd., Tokyo, Japan) images of the corresponding microstructure are shown in Figure 1 and Figure 2.

As shown in Figure 1, the peaks at 2920 cm^−1^, 2849 cm^−1^, 1596 cm^−1^, 1456 cm^−1^, 1375 cm^−1^, 1030 cm^−1^ and 810 cm^−1^ represent asymmetric stretching vibrations of –CH_2_–, symmetric stretching vibrations of –CH_2_–, stretching of C=C, scissor vibration of –CH_2_–, deformation vibration of –CH_3_, stretching vibration of S=O and =C–H out-plane deformation vibration of the vinyl group, respectively. Figure 2 shows that the surface of the base asphalt is smooth, and the base asphalt is, in general, a homogeneous structure.

#### 2.1.2. Crumb Rubber

In this study, crumb rubbers were produced from waste radial car tires. The mean diameters of untreated and plasticized crumb rubber particles were determined to be 0.25 mm (60 mesh) by screening, and the plasticized crumb rubber was produced by a constant pressure plasticizing method. The Fourier transform infrared spectra of the untreated and plasticized crumb rubbers are shown in Figure 3, and SEM images of the corresponding microstructures are shown in Figure 4.

The peaks in the untreated and plasticized crumb rubber spectra in Figure 3 only differ in intensity and not position. The intensity of the C=C stretching vibration at 1537 cm^−1^ of the plasticized crumb rubber is lower than that of the untreated crumb rubber, showing the partial cleavage of the C=C bond by plasticization [13]. However, the deformation vibration of the methyl group –CH_3_ at 1371 cm^−1^ is stronger for the plasticized crumb rubber than for the untreated crumb rubber, which indicates a reduction in the internal cross-linking of plasticized crumb rubber molecules. The increase in the intensity of the C–O stretching vibration at 1075 cm^−1^ and the S=O stretching vibration at 1030 cm^−1^ for the plasticized crumb rubber compared to the untreated crumb rubber reflects the oxidative fracture of C–S and S–S bonds during plasticization.

Figure 4 shows that the crumb rubber has a rough surface and a large specific surface area, which increases the contact between the crumb rubber and the base asphalt, whereas the plasticized crumb rubber has a smoother surface. The plasticized crumb rubber also has a relatively loose flocculent structure with a large number of closely spaced surface voids.

#### 2.1.3. Additive

The additive used in this study was Vestenamer^®^ 8012, which was supplied by Evonik Corporation USA (Parsippany, NJ, USA) Vestenamer^®^ 8012, is a trans-polyoctenamer rubber used to improve the chemical bond between the crumb rubber and the base asphalt, and the properties of this additive are listed in Table 3.

### 2.2. Preparation and Aging of Asphalt Rubbers

#### 2.2.1. Preparation of Asphalt Rubbers

The total crumb rubber content was 20% by weight of the base asphalt; different asphalt rubbers were prepared containing untreated crumb rubber contents of 20%, 10% and 0% (corresponding to 0%, 10% and 20% plasticized crumb rubber). The blended additive was 5% of the total mass of the crumb rubber. Three asphalt rubber samples were labeled as follows: 1# contained 20% untreated crumb rubber; 2# contained 10% untreated crumb rubber and 10% plasticized crumb rubber; and 3# contained 20% of plasticized crumb rubber. The following preparation process was used: First, 400 g of the base asphalt were weighed, placed in a cylindrical stainless steel container and heated at 180 °C. The two 60-mesh crumb rubbers and the additive were added to the SK 90# base asphalt and mixed at 180 °C at a stirring rate of 500 rpm. Then, the asphalt rubbers were stirred at 5000 rpm for 45 min at 180 °C by a high-speed shear dispersion emulsifying machine. Finally, the three asphalt rubbers were developed at 180 °C for 45 min and further tested.

#### 2.2.2. Aging of Asphalt Rubbers

The standard rolling thin film oven test (RTFOT) was implemented following ASTM D2872 [29] to simulate short-term aging of the asphalt rubbers. Then, the RTFOT residues of the asphalt rubbers were subjected to long-term aging following the standard pressurized aging vessel (PAV) method (ASTM D6521) [30] at 100 °C under an air pressure of 2.10 MPa for 20 h (1PAV) and 40 h (2PAV).

### 2.3. Rheological Test of Asphalt Rubbers

A dynamic shear rheometer (DSR) was used to conduct frequency sweep (FS) tests to evaluate the asphalt rubber crack resistance at low temperatures. The frequency range was 0.1–100 rad/s, and the strain was 0.1% at temperatures of PG + 10 °C and PG + 20 °C (PG: performance grade). The diameter of the DSR rotor was 4 mm, and the thickness of the samples was 2 mm.

### 2.4. Microscopic Test Techniques

#### 2.4.1. Fourier Transform Infrared Spectroscopy (FTIR)

The Fourier transform infrared spectra of the three asphalt rubbers were obtained by using attenuated total reflection. The depth of the resulting infrared spectrogram is generally a few microns, such that this surface analysis technology can be used to perform rapid, non-destructive and in situ detection. The detection wavenumber range ranged from 4000 cm^−1^ to 400 cm^−1^, and the peaks of functional groups in the spectra were subsequently analyzed.

#### 2.4.2. Scanning Electron Microscope (SEM)

A scanning electron microscope was used to observe the morphology of fractured cross-sections of the asphalt rubbers. The asphalt rubbers samples were coated with a gold thin film for the SEM tests.

### 2.5. Interconversion between Linear Viscoelastic Material Functions

In the developed DSR test specification, the reference temperature of the relaxation modulus master curve is increased from the low-temperature PG (performance grade) temperature by 10 °C to be consistent with the BBR test temperature, such that the viscoelastic mechanical response obtained in the frequency domain can be mapped to the viscoelastic properties obtained in the time domain. The evaluation indexes of the relaxation modulus G(60s) and the relaxation rate $m_{r}(60s)$ exhibited a good linear correlation with the creep stiffness modulus S(60s) and the creep rate m(60s) [31,32]. The interconversion between the linear viscoelastic material functions developed by Christensen (given in Equation (1) below [33]) was used to convert the modulus from the frequency domain to the time domain:(1)$$G(t)\approx {G^{\prime }(\omega )|}_{\omega =2/\pi t}$$
where G(t) is the relaxation modulus, MPa, $G^{\prime }(\omega )$ is the storage modulus, MPa, ω is the angular frequency, rad/s, and t is the loading time, s.

## 3. Results and Discussion

### 3.1. Master Curves of Storage Modulus for Different Aging Levels

As the PG temperatures of the three asphalt rubbers were −28 °C, the frequency sweep test temperatures were −18 °C and −8 °C and the reference temperature was −18 °C. The master curves of the storage modulus for different aging levels were plotted using the time−temperature equivalence principle. Figure 5 shows the master curves of the storage modulus that were converted into the relaxation modulus.

In Figure 5, similar change trends are observed for the storage moduli, that is, the storage modulus increases with the plasticized crumb rubber content. This result is obtained because the cross-links in the plasticized crumb rubbers are cleaved by atmospheric pressure plasticization—the molecular chains are shortened, and the number of active groups at the end of the molecular chains increases, thereby strengthening the bond between the crumb rubber and the polar groups in the base asphalt. Compared with the storage moduli of 1#, the storage moduli of 2# increase by at least 1.33% at 0.1 rad/s and 0.65% at 100 rad/s, whereas the storage moduli of 3# increase by at least 1.98% at 0.1 rad/s and 1.04% at 100 rad/s for the three aging levels. The storage modulus increases with the aging level, which promotes deformation recovery. Table 4 shows the fitting functions of the storage modulus, for which the coefficients of determination are all greater than 0.99.

### 3.2. Master Curves of Relaxation Modulus for Different Aging Levels

Equation (1) was used to convert the storage moduli $G^{’}(\omega )$ in Figure 5 to the relaxation moduli for the three asphalt rubbers for the different aging levels; the master curves for the relaxation modulus are displayed in Figure 6.

Figure 6 shows that the relaxation modulus increases with the plasticized crumb content. Compared with the storage moduli of 1#, the relaxation moduli of 2# increase by at least 1.05% at 0.1 rad/s and 0.82% at 100 rad/s, whereas the relaxation moduli of 3# increase by at least 1.69% at 0.1 rad/s and 1.30% at 100 rad/s for the three aging levels. Aging causes a considerable increase in the relaxation modulus. There is a decrease in the relaxation modulus, going from the 2PAV result to the 1PAV result to the RTFOT result. Table 5 shows the fitting functions of the relaxation modulus, for which the coefficients of determination are all greater than 0.99.

### 3.3. Converted Evaluation Index of Low-Temperature Rheological Properties

Two criteria must be satisfied for the BBR test at 60 s of loading time. The creep stiffness modulus should be smaller than 300 MPa, and the m-value should be larger than 0.300. The smaller the creep stiffness or the larger the m-value, the higher the low-temperature performance. Figure 7 and Figure 8 show the relaxation modulus G(60s) and the relaxation rate $m_{r}(60s)$ as converted evaluation indexes of the low-temperature rheological properties.

Although aging significantly increases the relaxation modulus G(60s), the relaxation modulus G(60s) is less than 300 MPa, and the relaxation rate $m_{r}(60s)$ is greater than 0.300 in Figure 7 and Figure 8. Aging has little effect on the relaxation rate $m_{r}(60s)$ of 1# and 2#, and the relaxation rate $m_{r}(60s)$ of 3# decreases as the aging level increases, which shows that aging decreases the low-temperature relaxation ability of asphalt rubber. The plasticized crumb rubber content significantly affects the relaxation modulus G(60s). The largest relaxation moduli are observed for 3# (which contains 20% plasticized crumb rubber) for the three aging levels, whereas the plasticized crumb rubber content has no clear effect on the relaxation rate $m_{r}(60s)$. The maximum differences in the relaxation rates of 1#, 2# and 3# are 6.13%, 4.89% and 7.86%, respectively. Additionally, an analysis of variance was employed to evaluate the influence of crumb rubber contents and aging on the low-temperature performance of asphalt rubber statistically. The standard deviations of relaxation moduli are 31.466 MPa, 42.453 MPa and 58.674 MPa for 1#, 2# and 3#, respectively. Accordingly, the standard deviations of relaxation rates are 0.006, 0.008 and 0.017 for 1#, 2# and 3#, respectively. The above statistical analysis indicates that the differences in low-temperature performance of asphalt rubbers caused by different aging levels become more significant with the increase in plasticized crumb rubber content.

### 3.4. FTIR

Table 6 lists the main functional groups used to identify the changes in the molecular structure of the asphalt rubbers with different crumb rubber and plasticized crumb rubber contents for the three aging levels [34,35,36,37], and the intensities of the absorption peaks at 1696 cm^−1^, 1596 cm^−1^ and 1030 cm^−1^ are listed in Table 7. Figure 9 shows the infrared spectra of three asphalt rubbers.

There is no clear difference between the FTIR spectra of the asphalt rubber and the base asphalt in Figure 9, showing that the ordinary and plasticized crumb rubbers are physically blended with the base asphalt, and the absorption of the lighter components by the crumb rubber particles produces a swelling reaction. The four strong absorption peaks at 2918 cm^−1^, 2849 cm^−1^, 1456 cm^−1^ and 1375 cm^−1^ for 1#, 2# and 3# correspond to the asymmetric stretching, symmetric stretching and scissor vibrations of –CH_2_– and the deformation vibration of –CH_3_, respectively. These vibrations are unchanged by adding crumb rubber to the base asphalt. The absorption peaks between 1000 cm^−1^ and 650 cm^−1^ that represent the C–C skeleton vibration and out-of-plane bending of C-H remain fairly consistent with the values given in Table 7.

However, there are significant differences among the three samples in the intensities of the absorption peaks at 1696 cm^−1^, 1596 cm^−1^ and 1030 cm^−1^ that correspond to C=O stretching, C=C stretching and the S=O stretching vibration, respectively. The strongest peaks are, in general, under 2PAV for 1#, 2# and 3#, and the peak intensities under 1PAV are greater than those of under RTFOT, which is attributed to 2PAV with longer aging oxidation time. The peak intensities for 3# under 2PAV and 1PAV are approximately equal and higher than under RTFOT. The highest peak intensities of all the samples are obtained for 3# under 2PAV, followed by sample 2#. Considering that 2# and 3# contain plasticized crumb rubber, parts of the molecular chains, such as C–S and S–S, are oxidized and broken during plasticization, which is easier to combine with oxygen after aging. Consequently, strong absorption peaks of C=O at 1696 cm^−1^ and S=O at 1030 cm^−1^ appear, especially for S=O of 3# at 1030 cm^−1^.

### 3.5. SEM

Scanning electron microscopy (SEM) was used to observe and analyze the microscopic morphology of the asphalt rubbers containing heterogeneous crumb rubber and the bonding between the crumb rubber and the base asphalt for the three aging levels. The micrographs are presented in Figure 10.

Figure 10 shows that the bonds between the crumb rubber and the base asphalt deteriorate as the aging level increases. The asphalt coating of the crumb rubbers deteriorates, the crumb rubber gradually separates from the base asphalt, and the interface between the base asphalt and crumb rubber becomes more distinct, showing that aging reduces the efficacy of the blending between the crumb rubber and the base asphalt and the performance of the asphalt rubber. This result is obtained because aging converts the light components in the base asphalt into macromolecular resins and asphaltenes that are weakly compatible with crumb rubbers, thereby weakening the interfacial interaction between the crumb rubber and the base asphalt.

The interface stripping degree of 1# is less than those of 2# and 3#, that is, the particle core of 1# remains more intact than for the other two samples. Aging gradually strips fine particles off the crumb rubber particles in 2# and 3#, especially for 3# under 2PAV. These stripped fine crumb rubber particles destroy the cross-linked network formed by the crumb rubber and the base asphalt. This result is obtained because molecular chains are broken in the plasticized crumb rubber in 2# and 3#. Aging destroys the interaction between the plasticized crumb rubber and the base asphalt, making it difficult for the plasticized crumb rubber to form particle cores. However, the crumb rubber in 1# is untreated and therefore sustains less damage under aging than the plasticized crumb rubber, such that the interfacial bonding between the untreated crumb rubber and the base asphalt remains relatively intact.

## 4. Conclusions

The impact of different aging levels on the low-temperature rheological properties of asphalt rubber prepared using untreated and plasticized crumb rubbers was investigated in this study. The evolution of the chemical groups and interfacial fusion between the base asphalt and the crumb rubbers were analyzed for asphalt rubbers containing heterogeneous crumb rubbers for different aging levels. The following conclusions were drawn from the rheological properties and microscopic characterization of the samples used in the study:

(1) The storage moduli of asphalt rubbers containing heterogeneous crumb rubbers increased with the plasticized rubber content and the aging level. An internal conversion relationship between the linear viscoelastic functions was used to obtain a changing trend for the converted relaxation modulus that was consistent with that of the storage modulus. Increasing the plasticized crumb rubber content and the aging level decreased the relaxation rate and the low-temperature performance of the asphalt rubber.

(2) No new absorption peaks appeared after adding the untreated and plasticized crumb rubbers to the base asphalt, showing that the crumb rubber was physically blended with the asphalt. Aging significantly affected the C=O, C=C and S=O absorption peaks, where the highest peak intensities were obtained for the sample containing only untreated crumb rubber. Low peak intensities were observed for the samples containing plasticized crumb rubber because plasticization produced broken molecular chains.

(3) The bonding between the crumb rubber and the base asphalt deteriorated as the aging level increased: the crumb rubbers gradually separated from the base asphalt, and the interface between the crumb rubber and the base asphalt gradually became distinct. Fine particles were stripped off the plasticized crumb rubbers, making particle core formation difficult, whereas the particle cores of the asphalt rubbers containing untreated crumb rubbers remained relatively intact.

Previous studies have shown that Vestanamer can improve the compatibility of crumb rubber and asphalt, and improve the storage stability of asphalt rubber. This paper investigated the low-temperature performance of asphalt rubber prepared by adding Vestanamer, and the low-temperature performance indexes met the specification requirements. The further research could consider adding a comparison group of asphalt rubber prepared without Vestanamer to compare and analyze the effect of Vestanamer on the low-temperature performance of asphalt rubber.

## Figures and Tables

**Figure 1 materials-13-04120-f001:**
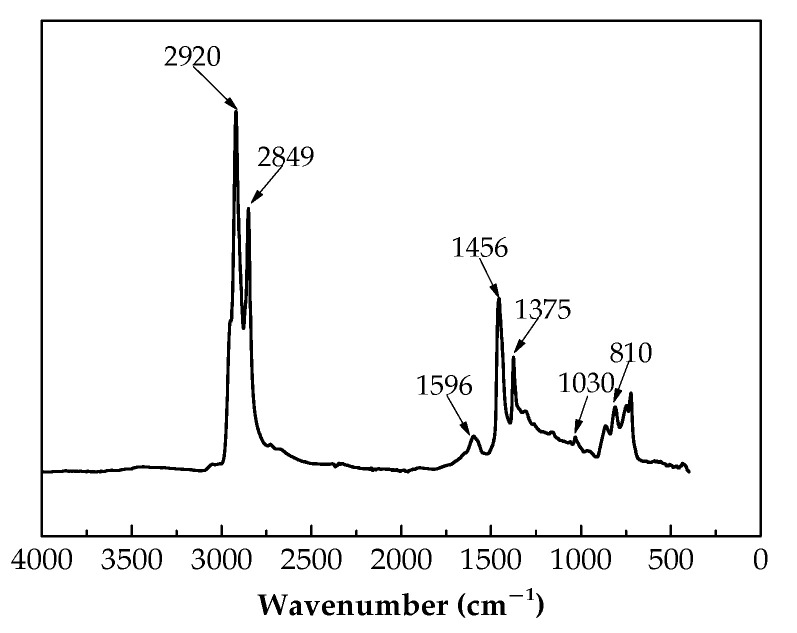
FTIR spectrum of the base asphalt.

**Figure 2 materials-13-04120-f002:**
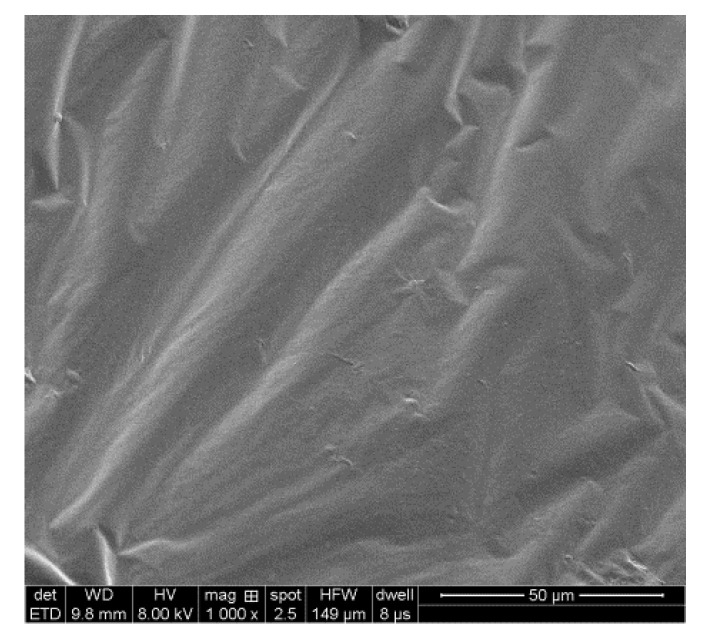
SEM image of the base asphalt.

**Figure 3 materials-13-04120-f003:**
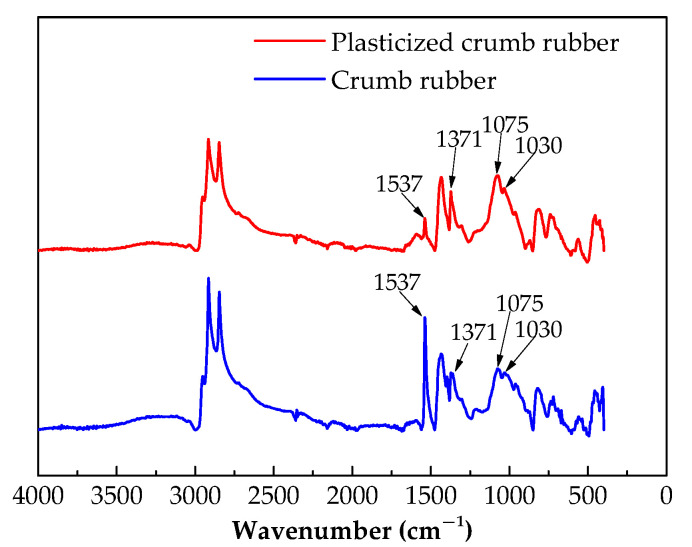
FTIR spectra of the untreated and plasticized crumb rubbers.

**Figure 4 materials-13-04120-f004:**
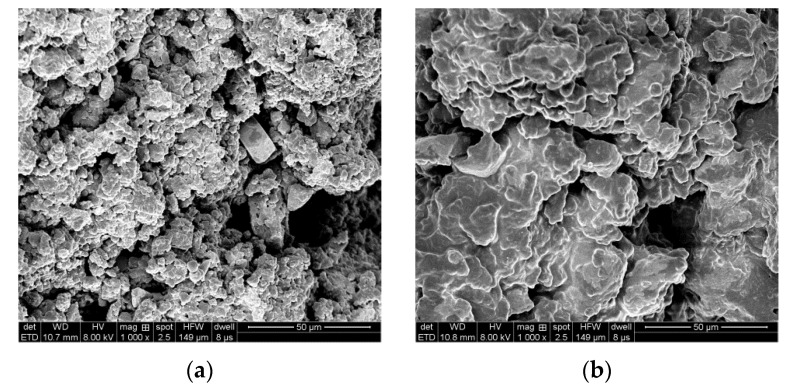
SEM images of crumb rubbers and plasticized crumb rubbers. (**a**) Crumb rubbers; (**b**) plasticized crumb rubbers.

**Figure 5 materials-13-04120-f005:**
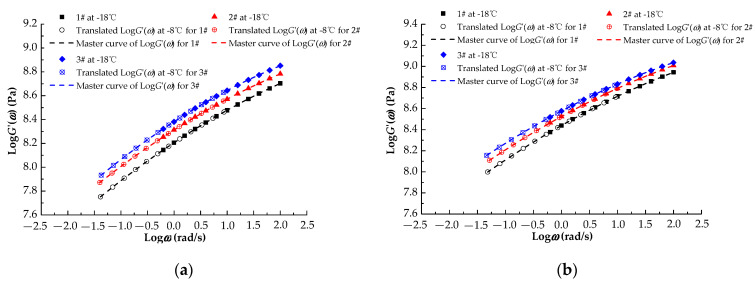
Master curves of storage modulus $G^{\prime }(\omega )$ for different aging levels. (**a**) RTFOT; (**b**) 1 pressurized aging vessel (PAV); (**c**) 2PAV.

**Figure 6 materials-13-04120-f006:**
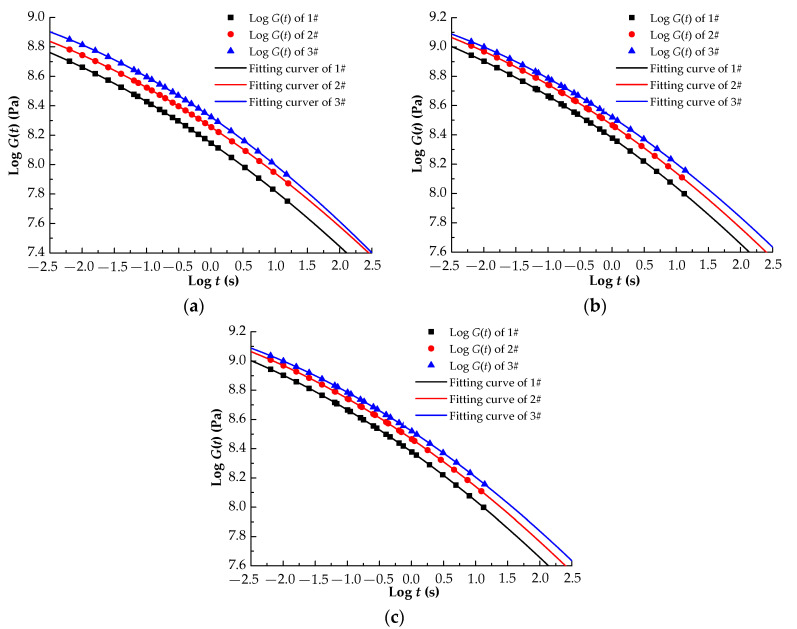
Master curves of relaxation modulus G(t) for different aging levels. (**a**) RTFOT; (**b**) 1PAV; (**c**) 2PAV.

**Figure 7 materials-13-04120-f007:**
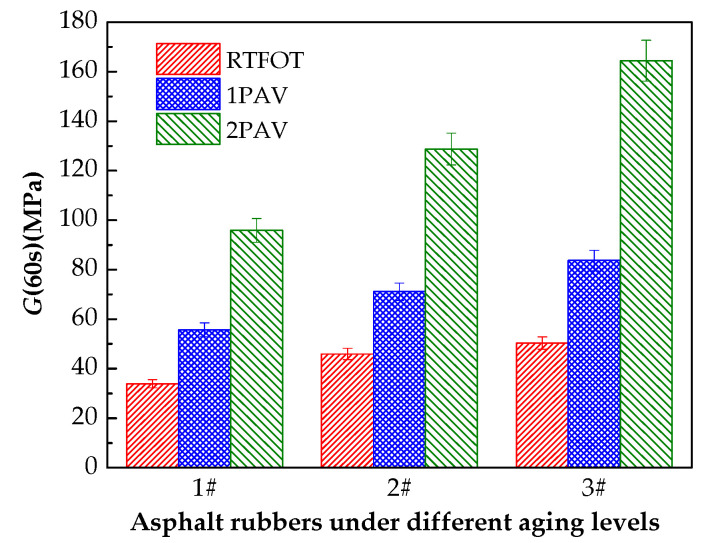
G(60s) for different aging levels.

**Figure 8 materials-13-04120-f008:**
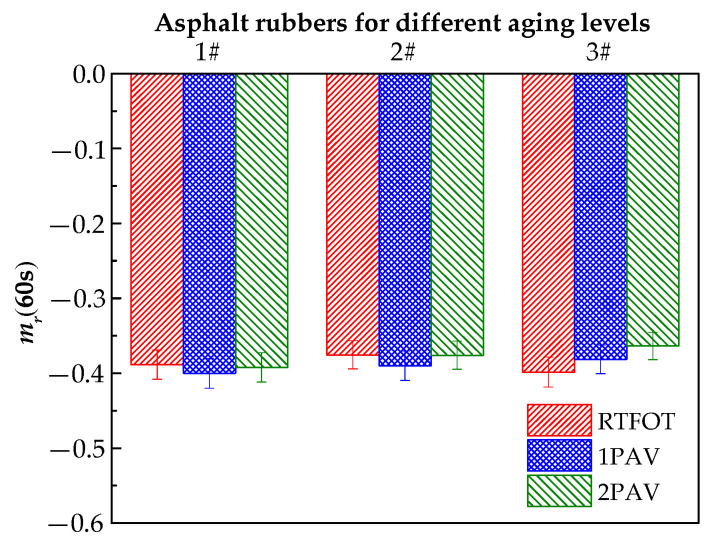
$m_{r}(60s)$ for different aging levels.

**Figure 9 materials-13-04120-f009:**
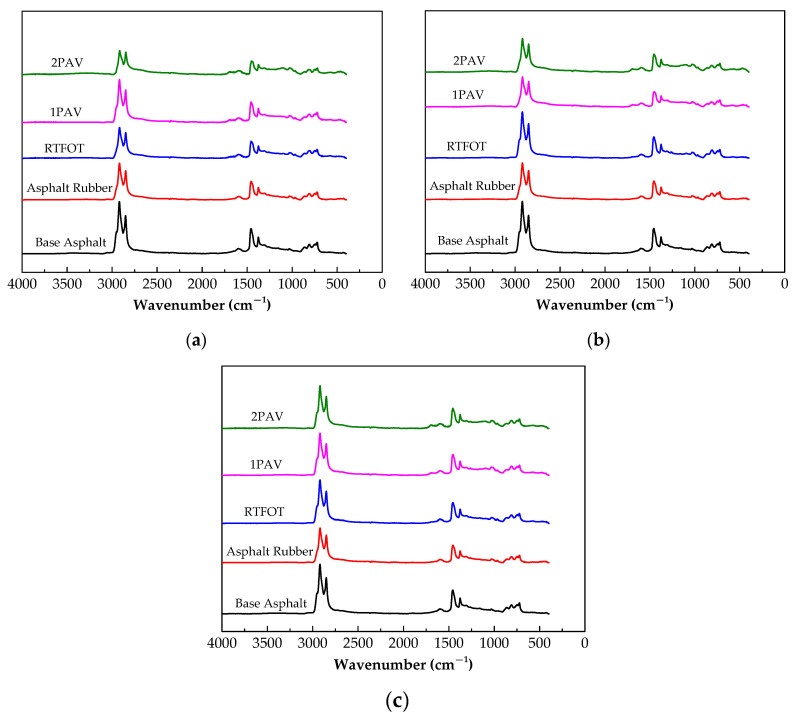
FTIR spectra of asphalt rubbers for different aging levels. (**a**) 1#; (**b**) 2#; (**c**) 3#.

**Figure 10 materials-13-04120-f010:**
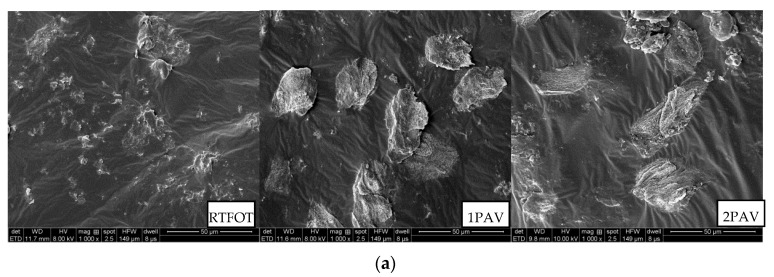
SEM images of asphalt rubbers for different aging levels. (**a**) 1#; (**b**) 2#; (**c**) 3#.

**Table 1 materials-13-04120-t001:** Basic property indexes of base asphalt.

Property Indexes		Tested Results
Penetration (25 °C, 5 s, 100 g, 0.1 mm)		89.8
Softening point (°C)		44.5
Ductility (15 °C, 5 cm/min, cm)		>100
RTFOT	Residual penetration ratio (%)	54.01
	Quality change (%)	0.4
Apparent viscosity (135 °C, Pa·s)		0.594

Notes: RTFOT represents rolling thin film oven test.

**Table 2 materials-13-04120-t002:** Compositions of base asphalt.

Compositions	Tested Results
Saturates (%)	16.67
Aromatics (%)	43.44
Resins (%)	26.01
Asphaltenes (%)	13.88

**Table 3 materials-13-04120-t003:** Basic properties of Vestenamer^®^ 8012.

Items	Tested Results
Physical state	solid
Color	white
Form	granular
Glass transition temperature (°C)	−65
Melting point (°C)	54
Density (g/cm^3^)	0.91

**Table 4 materials-13-04120-t004:** Fitting parameters of master curves of storage modulus for the three aging levels.

Aging Levels		Fitting Functions of Storage Modulus y = ax^2^ + bx + c		
		a	b	c
RTFOT	1#	−0.02361	0.29532	8.20603
	2#	−0.02361	0.2822	8.31233
	3#	−0.0276	0.2894	8.38175
1PAV	1#	−0.02489	0.30172	8.43949
	2#	−0.02497	0.29138	8.52523
	3#	−0.02553	0.2806	8.577
2PAV	1#	−0.02459	0.29507	8.66109
	2#	−0.02481	0.27793	8.75594
	3#	−0.02443	0.26712	8.83934

**Table 5 materials-13-04120-t005:** Fitting parameters of master curves of relaxation modulus for the three aging levels.

Aging Levels		Fitting Functions of Relaxation Modulus y = ax^2^ + bx + c		
		a	b	c
RTFOT	1#	−0.02361	−0.30458	8.1472
	2#	−0.0236	−0.2914	8.25608
	3#	−0.0276	−0.30025	8.32392
1PAV	1#	−0.02489	−0.31148	8.37936
	2#	−0.02497	−0.30117	8.46712
	3#	−0.02553	−0.29061	8.52099
2PAV	1#	−0.02459	−0.30472	8.60228
	2#	−0.02481	−0.28766	8.70448
	3#	−0.02443	−0.2767	8.78601

**Table 6 materials-13-04120-t006:** Functional groups in asphalt rubbers [34,35,36,37].

Wavenumber (cm^−1^)	Functional Groups
2918	asymmetric stretching vibrations of –CH_2_–
2849	symmetric stretching vibrations of –CH_2_–
1696	stretching of C=O
1596	stretching of C=C
1456	scissor vibration of –CH_2_–
1375	deformation vibration of –CH_3_
1105	stretching vibration of C–O
1030	stretching vibration of S=O
650–1000	substitution region of benzene ring

**Table 7 materials-13-04120-t007:** Intensities of three absorption peaks.

Wavenumber (cm^−1^)	Aging Levels	Intensities		
		1#	2#	3#
1696	RTFOT	0.00818	0.00583	0.0083
	1PAV	0.01476	0.01117	0.01387
	2PAV	0.01435	0.0175	0.01995
1596	RTFOT	0.02042	0.01959	0.02305
	1PAV	0.02443	0.01982	0.02546
	2PAV	0.02087	0.02419	0.02724
1030	RTFOT	0.02962	0.02972	0.03184
	1PAV	0.03514	0.02714	0.04585
	2PAV	0.03604	0.04153	0.04475

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
