# Peer review of "Low-Temperature Rheological Properties and Microscopic Characterization of Asphalt Rubbers Containing Heterogeneous Crumb Rubbers"

_materials, 2020, doi:10.3390/ma13184120_

Round 1

Reviewer 1 Report

Low Temperature Rheological….

The paper analyses the use of the rubber crumbs derived from grinding waste tires in the preparation of bituminous composition for asphalt.

The reviewer has for the authors the following questions:

  • Why the authors have forgotten to identify which type of the recycled tires are used? The reviewer remembers to the authors that the tire producers use their own recipes in which the rubbers are different and in different ratios. The items performances are different in terms of mechanical-elastic characteristics and can have different responses in terms of safety, economy and comfort. It is important also to consider the type of vulcanization, the ratio between the rubbers used and the additives linked with the final performances. This last subject is exemplified in part in the huge numerical work made by other scholar's research groups and should be mentioned, analyzed and acknowledged properly, for example:
  •    * Math. Chem., 52(2), 464 – 2015
  • In particular, it is necessary to verify if the different types of rubbers, their ratios and their recipes have an influence on crumbs used in asphalt compounds. Moreover the reviewer would like that the authors compare the performances of their experimental work with the results claimed in the US Patent N° 4829109.
  • The authors could add in the paper the table of the asphalt composition in phr.
  • In order to be completely clear, the authors shall add for the compounds the following determinations:
  • The penetration at 25 and 50°C following the ASTM D 5 method
  • The Brookfield viscosity following the UNI ISO 2555:2002 method
  • The linear thermal dilatation following the UNI 8202 method
  • The drainage capacity of the asphalt compound.

provided that the previous issues are resolved the paper is acceptable for publication

Author Response

Point 1: Why the authors have forgotten to identify which type of the recycled tires are used? The reviewer remembers to the authors that the tire producers use their own recipes in which the rubbers are different and in different ratios. The items performances are different in terms of mechanical-elastic characteristics and can have different responses in terms of safety, economy and comfort. It is important also to consider the type of vulcanization, the ratio between the rubbers used and the additives linked with the final performances. This last subject is exemplified in part in the huge numerical work made by other scholar's research groups and should be mentioned, analyzed and acknowledged properly, for example:

   * Math. Chem., 52(2), 464 – 2015

In particular, it is necessary to verify if the different types of rubbers, their ratios and their recipes have an influence on crumbs used in asphalt compounds. Moreover the reviewer would like that the authors compare the performances of their experimental work with the results claimed in the US Patent N° 4829109.

Response 1: Thank you very much for your professional and careful comments. Tire compositions affect the properties of crumb rubber and then affect the performance of asphalt rubber. The descriptions are added in lines 42 and 43, and two references have been supplemented in "References".

The recycled tires used in this paper are waste radial car tires, and the corresponding content is added in line 109. We compare the experimental results in this paper with the results in the US Patent N° 4829109. And we find that the polymeric compositions (polyethylene, polypropylene) were used as asphalt modifiers and the tested index was the Mooney viscosity in the US Patent N° 4829109 (Mooney viscosity is a high temperature index). However, untreated and plasticized crumb rubbers were used as asphalt modifiers and the research object was the low-temperature properties in this paper. So, we think a comparative analysis can be carried out in the follow-up study on high-temperature performance of asphalt rubber.

Point 2: The authors could add in the paper the table of the asphalt composition in phr.

In order to be completely clear, the authors shall add for the compounds the following determinations:

The penetration at 25 and 50°C following the ASTM D 5 method

The Brookfield viscosity following the UNI ISO 2555:2002 method

The linear thermal dilatation following the UNI 8202 method

The drainage capacity of the asphalt compound.

Response 2: Thank you very much for your valuable comments. In the Revised manuscript, the compositions of base asphalt are supplemented in Table 2.

The penetration at 25 °C and Brookfield viscosity are listed in the following Table.

Tested results of the penetration and Brookfield viscosity

Property Indexes

Tested Results

1#

2#

3#

Penetration (25°C, 5s, 100g, 0.1mm)

29.5

38.1

41.9

Brookfield viscosity (Pa·s)

135 °C

8.933

4.432

3.833

180 °C

1.983

1.223

0.838

The penetration, Brookfield viscosity and linear thermal dilatation are the high temperature performance indexes. And the research object is the low-temperature properties in this paper, so the penetration, Brookfield viscosity, linear thermal dilatation and drainage capacity have no correlation with the low temperature performance of asphalt rubber. If it is necessary, we can add these tests in the follow-up study on high-temperature performance of asphalt rubber. Finally, we still thank the reviewer for your careful comments.

Reviewer 2 Report

The writing in the first sentence in the Introduction is odd, please revise.

Although some readers may be familiar with FTIR and SEM images, it would be worth explaining or briefly describing what is shown in Figures 1 and 2. The authors did a very good job describing Figures 3 and 4.

In section 3.3, a statistical analysis such as ANOVA or GLM would help support the statements made (whether the effect of adding rubber is significant or not). The conclusions made seem to be purely based on visual observations.

The results shown in Figure 9 can also be supported with a table, since it is not possible to read the intensities of the FTIR spectra. Lines 247-259 discuss significant differences in the intensities of the absortion peaks at 1696, 1596 and 1030 cm-1; these results can be shown in a table to provide better information to the reader.

Author Response

Point 1: The writing in the first sentence in the Introduction is odd, please revise.

Response 1: Thank you very much for your helpful comment. The revised sentence is supplemented in line 39.

Point 2: Although some readers may be familiar with FTIR and SEM images, it would be worth explaining or briefly describing what is shown in Figures 1 and 2. The authors did a very good job describing Figures 3 and 4.

Response 2: Thank you very much for your valuable comment. We add the explaining and describing what is shown in Figures 1 and 2 in lines 103-107.

Point 3: In section 3.3, a statistical analysis such as ANOVA or GLM would help support the statements made (whether the effect of adding rubber is significant or not). The conclusions made seem to be purely based on visual observations.

Response 3: Thank you very much for your professional and careful comment. An analysis of variance is provided in lines 242-248 of "3.3. Converted Evaluation Index of Low-Temperature Rheological Properties ".

Point 4: The results shown in Figure 9 can also be supported with a table, since it is not possible to read the intensities of the FTIR spectra. Lines 247-259 discuss significant differences in the intensities of the absortion peaks at 1696, 1596 and 1030 cm-1; these results can be shown in a table to provide better information to the reader.

Response 4: Thank you very much for your valuable and careful comment. The intensities of the absortion peaks at 1696, 1596 and 1030 cm-1 are listed in Table 8 and the analysis on absorption peaks is revised in "3.4. FTIR".

Reviewer 3 Report

The article deals with an important and interesting topic. Rubber modification is still a valid issue. The authors used advanced methods to assess the influence of the modification method on the properties of asphalt. The article is well written and easy to read. My comments:
1) In the literature review, I propose to add some important conclusions from the cited works, which are related to the topic of your research. The current text basically only indicates who did what, no significant conclusions.
2) Is the base asphalt plain or modified?
3) Was there no similar research with Vestanamer in the literature - this is a measure known for many years. It is worth mentioning something about this in the literature.
4) Figures 7 and 8 require commentary regarding the statistical significance of the differences.
5) The results of using Vestanamer are generally not optimistic, if it can be broadly commented on in the summary. Whether to apply or not. May indicate an agenda for further research.

Author Response

Point 1: In the literature review, I propose to add some important conclusions from the cited works, which are related to the topic of your research. The current text basically only indicates who did what, no significant conclusions.

Response 1: Thank you very much for your valuable comments. We add some conclusions from the cited works in lines 75-81.

Point 2: Is the base asphalt plain or modified?

Response 2: The base asphalt is plain.

Point 3: Was there no similar research with Vestanamer in the literature - this is a measure known for many years. It is worth mentioning something about this in the literature.

Response 3: Thank you very much for your helpful comment. There are some researches on asphalt rubber with Vestanamer and the researches are added in lines 72-74 and "References".

Point 4: Figures 7 and 8 require commentary regarding the statistical significance of the differences.

Response 4: Thank you very much for your careful comment. An analysis of variance and the statistical significance of the differences are provided in lines 242-248.

Point 5: The results of using Vestanamer are generally not optimistic, if it can be broadly commented on in the summary. Whether to apply or not. May indicate an agenda for further research.

Response 5: Thank you very much for your valuable comments. According to the Reference 25 (Yadollahi, G.; Mollahosseini H. S.), the main role of Vestenamer is to improve the performance of pavement at high temperature conditions. Previous studies have shown that Vestanamer can improve the compatibility of crumb rubber and asphalt, and improve the storage stability of asphalt rubber. This paper studies the low temperature performance of asphalt rubber prepared by adding Vestanamer, and the low temperature performance indexes meet the specification requirements. The further research can consider adding a comparison group of asphalt rubber prepared without Vestanamer to compare and analyze the effect of Vestanamer on the low temperature performance of asphalt rubber. The comments are added in lines 337-342.
